# Isolation and Identification of Bacteria *Bacillus velezensis* with Antagonistic Activity Against Major Pathogens of Coconut

**DOI:** 10.3390/microorganisms13112640

**Published:** 2025-11-20

**Authors:** Hui Zhu, Sajid Mehmood, Xiaoqing Niu

**Affiliations:** 1Coconut Research Institute of Chinese Academy of Tropical Agricultural Sciences, Wenchang 571339, China; zhuhui@catas.cn; 2Hainan Innovation Center of Academician Team, Wenchang 571339, China; 3Department of Plant Pathology, Faculty of Agriculture, Pir Mehr Ali Shah Arid Agriculture University, Rawalpindi 46000, Pakistan

**Keywords:** coconut, *Bacillus velezensis*, biological control, disease management, China

## Abstract

Coconut (*Cocos nucifera* L.) cultivation is increasingly threatened by destructive fungal pathogens that reduce yield and compromise tree health, particularly in tropical regions in China. To address this challenge, the present study aimed to isolate and identify antagonistic bacteria with potential biocontrol activity against key pathogens of coconut, including *Ceratocystis paradoxa* (crown rot), *Pestalotiopsis menezesiana* (leaf blight), and *Curvularia oryzae* (leaf spot). A total of 65 bacterial strains were isolated from 58 soil samples collected from healthy coconut orchards. Among these, strain X1 exhibited the most vigorous antagonistic activity, with inhibition rates exceeding 70% against multiple pathogens in both plate confrontation and mycelial growth rate assays. Morphological, physiological, biochemical, and molecular (16S rDNA and *gyrB* gene sequencing) analyses confirmed the identity of strain X1 as *Bacillus velezensis*. Optimization of fermentation conditions for strain X1 revealed that maximum antimicrobial activity was achieved using a medium containing 2% glucose, 3% peptone, 0.3% NaH_2_PO_4_·2H_2_O, and 0.4% Na_2_HPO_4_·2H_2_O, at 28 °C and pH 7.0–7.5, 5% inoculum, 50–70 mL liquid volume in 250 mL flasks, 72 h fermentation, and agitation at 180–220 rpm. These findings highlight *B*. *velezensis* X1 as an up-and-coming biocontrol agent with dual functionalities: effective suppression of coconut fungal pathogens and potential plant growth promotion. Its application could significantly reduce the dependence on synthetic fungicides, offering an eco-friendly alternative for integrated disease management in coconut farming.

## 1. Introduction

Coconut (*Cocos nucifera* L.), a versatile and economically important crop, plays an important role in the agricultural economies of tropical and subtropical regions. It is a source of vital food, fuel, fiber, and oil, with millions of people depending on it for sustenance and livelihood [1]. However, coconut cultivation faces several challenges, primarily from a variety of pathogens that cause significant yield losses. Among these, lethal yellowing, caused by phytoplasmas, and bud rot, caused by fungi, are particularly destructive. Mycoplasma-like organisms responsible for lethal yellowing cause rapid yellowing and death of palms, severely reducing their productive lifespan. Additionally, the coconut scale insect and *Ganoderma lucidum*, a fungal pathogen causing basal stem rot, further compromise plant health [1]. These pathogens not only diminish productivity but also increase control costs, threatening the sustainability of coconut farming globally. Particularly, the fungal pathogen of coconut diseases, with their devastating impact, are particularly alarming., posing a significant threat to coconut cultivation in Hainan Province in China, impacting both yield and quality. Notably, *Ceratocystis paradoxa* causes stem bleeding, characterized by dark discoloration and exudation from infected trunks, leading to tree death within a few months. This disease severely hampers sustainable coconut production by reducing tree longevity and productivity. Additionally, fungal diseases such as leaf blight caused by *Pestalotiopsis menezesiana* and leaf spot caused by *Curvularia oryzae* further threaten young and mature plants, resulting in premature leaf shedding, reduced vigor, and early fruit drop. These pathogens can spread rapidly under favorable conditions, exacerbating economic losses. Proper management strategies, including vigilant cultural practices, wound prevention, and early diagnosis, are crucial to controlling these diseases. Implementing integrated disease management is essential for maintaining healthy coconut stands, ensuring sustainable production, and safeguarding the economic importance of coconut across Hainan [2,3,4].

Coconut pathogens pose significant threats to the productivity and longevity of coconut plantations worldwide. The management of coconut diseases has traditionally relied on the use of chemical pesticides and fungicides. However, the extensive application of these chemicals has led to environmental pollution, the development of pathogen resistance, and harm to non-target organisms [5]. The spread of these pathogens is exacerbated by the lack of effective control methods and the environmental constraints of tropical ecosystems [6]. Conventional management strategies, with their limited success and recurring outbreaks of disease even after pesticide treatments, underscore the need for more innovative approaches that address these pathogens in an environmentally sustainable manner. In recent years, there has been growing interest in the role of beneficial microorganisms, as the biological control agents (BCAs). BCAs have gained momentum as a sustainable and eco-friendly alternative. Notably, *Bacillus velezensis*, a species within the genus *Bacillus*, has recently emerged as a promising biocontrol agent against plant pathogens, including those affecting coconut plantations [7]. This bacterium produces various secondary metabolites, including antibiotics and lipopeptides, which are toxic to pathogens and inhibit their growth. Moreover, *B*. *velezensis* has demonstrated effective plant growth-promoting abilities, making it a dual-purpose agent for disease control and plant health enhancement [8].

The application of *B*. *velezensis* as a biocontrol agent involves production of antimicrobial compounds [9] such as fengycin, iturin, and surfactin, which directly inhibit the growth of pathogenic fungi and bacteria [10]. Additionally, *B*. *velezensis* can induce systemic resistance in watermelon against *Fusarium* wilt, enhancing their defense mechanisms against pathogen attacks [11]. As a result, this bacterium has been extensively studied for its potential use in controlling a range of coconut pathogens, including *Phytophthora palmivora* (leaf blight), *Ganoderma boninense* (basal stem rot), and *Fusarium* spp. (wilt diseases) [5,7]. Recent research indicates that *B*. *velezensis* is not only an effective antagonist against pathogens but also capable of promoting the growth of coconut plants under conditions of pathogen-induced stress. This dual functionality of *B*. *velezensis* can be utilized in integrated pest management (IPM) strategies for sustainable cultivation of coconut [12]. Applying BCAs may reduce the reliance on synthetic chemicals, encourage sustainable farming practices, and enhance the overall health and yield of coconut palms.

Research focusing on the isolation and identification of *B*. *velezensis* from different environmental sources, including soil and plant rhizospheres, has revealed its potential to act as a biocontrol agent against various coconut pathogens. The success of such approaches depends on identifying strains with high antagonistic activity, stability, and compatibility with the plant host [13]. Understanding the molecular mechanisms underlying the biocontrol properties of *B*. *velezensis* and its interactions with coconut pathogens are critical for developing effective biocontrol formulations. Metagenomics and genomic sequencing technologies have played a pivotal role in identifying these bacteria at the genetic level, thereby enabling the discovery of novel strains with enhanced antagonistic properties [8]. The *gyrB* gene, encoding the subunit B protein of DNA gyrase, is a practical tool for species identification in the *B*. *subtilis* group. This type II DNA topoisomerase, universally distributed among bacterial species, is a crucial element in DNA replication. The rate of molecular evolution inferred from *gyrB* gene sequences is notably faster than that inferred from 16S rRNA gene sequences. This gene has been a subject of interest in phylogenetic studies of *Pseudomonas*, *Acinetobacter*, *Mycobacterium*, *Salmonella*, *Shigella*, *Escherichia coli*, *Aeromonas*, and the *Bacillus anthracis*-*cereus*-*thuringiensis* group. The results from these studies have consistently pointed to *gyrB* as a reliable phylogenetic marker for studying species-level phylogenetic and taxonomic relationships. In our present study, we have demonstrated that direct sequencing of the *gyrB* gene is a powerful tool for the identification and phylogenetic analysis of species in the *B. subtilis* group [14].

This study aimed to isolate and identify antagonistic bacteria from different coconut-growing regions in Hainan Province and evaluate their antagonistic activity against major coconut pathogens. The findings from this research will not only contribute to the development of effective biocontrol solutions but also offer practical insights into the potential of *B*. *velezensis* as part of an integrated disease management strategy for coconut crops, making it a valuable resource for the agricultural community.

## 2. Materials and Methods

### 2.1. Soil Sampling

A total of 58 soil samples were collected at a depth of 25 to 35 cm below the ground of over 10-year-old healthy coconut trees along an “S” curve at each point. These points were specifically located in Wenchang (E110.779783, N19.552650), Qionghai (E110.779783, N19.552650), and Wanning (E110.366087, N18.692141) counties in Hainan Province, China. The samples were then brought to the laboratory and stored at 4 °C.

### 2.2. Test Pathogens

For this study, we used the previously isolated pathogens of *C*. *paradoxa*, causing stem basal rot of coconut (GenBank Accession No. HQ248205.1) [2]; *P*. *menezesiana* (GenBank Accession No. KJ605161) [3], causing leaf blight of coconut; and *C*. *oryzae*, causing leaf spot of coconut (GenBank Accession No. MN180224) [4]. These pathogens have been reported as a major threat to coconut plantations in Hainan, China [2,3,4].

### 2.3. Growth Media

Liquid broth (LB) (10 g tryptone, 5 g yeast extract, 5 g NaCl, 18–20 g agar, 1000 mL distilled water, pH 7.0). Nutrient agar (NA) beef extract peptone medium (3 g beef extract, 10 g peptone, 10 g NaCl, and 18 g agar, 1000 mL distilled water pH 7.2–7.4). This medium, in addition to isolating and culturing antagonistic bacteria, also plays a crucial role in preserving the strains, ensuring their viability for future studies.

Potato dextrose agar (PDA) 200 g/L potato, 20 g glucose, 17–20 g agar powder, 1000 mL of distilled water. LB seed media (10 g tryptone, 5 g yeast extract, 10 g NaCl, 17–20 g agar, 1000 mL distilled water, pH 7.5 1 mol L^−1^ NaOH. Basic fermentation was prepared using 2% glucose, 2% peptone, 0.05% MgSO_4_·7H_2_O, 0.2% NaH_2_PO_4_·2H_2_O, 0.4% Na_2_HPO_4_·2H_2_O, 0.02% CaCl_2_, at a pH 7.0–7.2.

Fermentation media (A). Gauze’s liquid medium No. 1 (20.0 g soluble starch, 1.0 g KNO_3_, 0.5 g NaCl, 0.5 g MgSO_4_·7H_2_O, 0.5 g K_2_HPO_4_, 10.0 mg FeSO_4_·7H_2_O, 1000 mL water, pH 7.4, 15.0 g agar added as needed); (B) glucose yeast extract broth (glucose 10 g/L, yeast extract 4 g/L, NaCl 1 g/L, KH_2_PO_4_ 1 g/L); (C). glucose peptone broth (glucose 10 g/L, peptone 10 g/L, CaCO_3_ 2 g/L, NaCl 2.5 g/L); (D) soybean meal liquid fermentation media using soybean meal 10 g/L, NaCl 2.5 g/L, CaCO_3_ 2 g/L, peptone 3 g/L, glucose 10 g/L; (E) Czapek media using NaNO_3_ 2 g/L, K_2_HPO_4_ 1 g/L, KCl 0.5 g/L, MgSO_4_·7H_2_O 0.5 g/L, FeSO_4_·7H_2_O 0.01 g/L, sucrose 30 g); (F) BPY media using glucose 5 g/L, beef extract 5 g/L, peptone 1 g/L, yeast powder 5 g/L, NaCl 5 g/L); (G) starch ammonium salt media using soluble starch 10 g/L, (NH_4_)_2_SO_4_ 2 g/L, K_2_HPO_4_ 1 g/L, MgSO_4_ 1 g/L, NaCl 1 g/L, CaCO_3_ 3 g/L); (H) corn flour broth by mixing corn flour 40 g/L, glucose 10 g/L, and peptone 10 g/L; (I) sucrose soluble starch broth was prepared by adding sucrose 15 g/L, soluble starch 15 g/L, soybean meal 5 g/L, yeast extract 10 g/L, and K_2_HPO_4_ 8 g/L; (J) potato dextrose broth (PDB) using potato 200 g/L, and glucose 20 g/L in 1000 mL of distilled water, respectively [15].

### 2.4. Chemicals and Reagents

We used a 5% hydrogen peroxide solution for the catalase test (Sigma-Aldrich, St. Louis, MO, USA). 5 g of iodine and 10 g of potassium iodide were dissolved in 85 mL of distilled water to perform a starch hydrolysis test (Thermo Fisher Scientific, Waltham, MA, USA). For the Voges–Proskauer (VP) test, a 5% α-naphthol solution and a 40% NaOH solution (Merck, Darmstadt, Germany) were carefully mixed and used immediately. 0.1 g of methyl red, dissolved in 300 mL of 95% ethanol and 200 mL of distilled water, was added for the methyl red test (Sigma-Aldrich, St. Louis, MO, USA). For the nitrate reduction test, 15 mL of diluted acetic acid was added to 0.5 g of p-amino benzenesulfonic acid and 0.1 g of α-naphthylamine (Thermo Fisher Scientific, Waltham, MA, USA), which were mixed with 20 mL of distilled water. A solution was prepared by mixing oxalic acid, crystal violet, 95% ethanol, iodine solution, and 0.5% safranin as a counterstain for the Gram staining test (Sigma-Aldrich, St. Louis, MO, USA).

### 2.5. Equipment and Instruments

Vertical pressure steam autoclave (LDZF-65KB, Shanghai Shen’an Medical Instrument Factory, Shanghai, China); vertical flow clean bench (HCB-1300V, Haier Group, Qingdao, China); constant temperature and humidity incubator (SPX-158, Ningbo Jiangnan Instrument Factory, Ningbo, China); refrigerator (BCD-258WDPM, Haier Group); ultrapure water machine (Pulifair FST, Shanghai Fushite Instrument Equipment Co., Ltd., Shanghai, China); microscope (Leica DMIL, Leica, Wetzlar, Germany); electric blast drying oven (DHG-9030A, Shanghai Yiheng Scientific Instrument Co., Ltd., Shanghai, China); small desktop centrifuge (Centrifuge 5418, Eppendorf, Hamburg, Germany); PCR instrument (TProfessional, Biometra, Göttingen, Germany); electrophoresis instrument (DYY-4C, Beijing Liuyi Instrument Factory, Beijing, China); vertical flow clean bench (HCB-1300V, Haier Group); gel imaging system (Tanon 3500, Shanghai Tianneng Technology Co., Ltd., Shanghai, China); UV analyzer (WD-9403D, Beijing Liuyi Instrument Factory, Beijing, China).

### 2.6. Isolation and Purification of Bacteria

Bacteria were isolated using a 10-fold incremental serial dilution method. The soil samples were air-dried and passed through a 60-mesh sieve. Each soil sample was diluted 10, 10^2^, 10^3^, and 10^4^ times. 0.1 mL of each gradient soil dilution was pipetted onto an LB culture plate and evenly spread with a stainless-steel coating rod. After drying, the Petri dishes were inverted and incubated at 25–28 °C for 48 h. Based on the color, morphology, and size of the colonies grown on the plate, a thorough selection process was undertaken to ensure the highest quality of research. Single colonies of different bacteria were meticulously chosen and transferred onto a new culture medium for purification. The purified strains were numbered and preserved for further studies.

### 2.7. Preliminary Screening of Antagonistic Bacteria

Antagonism tests of the isolated bacterial strains were conducted against three pathogens using a disk diffusion method [16]. A 7 mm diameter PDA plug of the target pathogen was inoculated in the center of a 90 mm diameter PDA plate. Strain X1 was applied to the PDA medium using an inoculum suspension prepared to a density of approximately 10^−8^ CFU/mL, and 100 µL of this suspension was inoculated onto the surface of the PDA medium. A 7 mm diameter bacterial plug was vertically inoculated 2.5 cm away from the target pathogen. Three of the four cross-symmetrical sites on the plate were inoculated with the same test strain as replicates, and one site was left uninoculated as a crucial control. After incubation at 25–28 °C for 48 h, the width of the inhibition zone between the test strain and the target strain was measured, serving as an indicator of the effectiveness of the antagonistic bacteria.

### 2.8. Evaluation of Antimicrobial Activity of Antagonistic Bacteria

According to the results of the plate confrontation test, we identified strains with strong antagonistic effects. Their antimicrobial effects were determined using the mycelial growth rate inhibition method [17]. The antagonistic bacteria were cultured in LB liquid medium at 28 °C and 180 rpm for 48 h, then centrifuged at 8000 rpm. The supernatant was filtered through a 0.22 μm sterile filter membrane and thoroughly mixed with 40–50 °C PDA culture medium at a volume ratio of 1:20. After the culture medium solidified, a 7 mm diameter mycelium block of the target strain was inoculated in the center of the plate, and the treatment of inoculating the target strain alone was used as a control. Each treatment was repeated three times, ensuring thorough data collection, and cultured in a constant-temperature culture room at 25–28 °C. When the colonies on the control culture dish almost covered the entire surface, the diameter of the colonies for each treatment was measured, and the inhibition zones were calculated using the following equation.$$Inhibition\;rate=\frac{Control\;diameter-Treated\;diameter}{Control\;diameter-cake\;diameter}\times 100\%$$

### 2.9. Characterization of Antagonistic Bacteria

#### 2.9.1. Morphological Identification

The antagonistic bacteria to be identified were transferred to plates for culturing. From the second day onwards, the colony morphology, colony texture, colony edge regularity, optical properties, colony color, and whether pigments could be secreted were observed daily. The microscopic morphological characteristics of the strains, which included the size, shape, color, and type of spore-producing structures, were of utmost importance in our research.

#### 2.9.2. Determination of Physiological and Biochemical Characteristics

Catalase test, starch hydrolysis test, Voges–Proskauer (VP) [18], methyl red test [19], hydrogen sulfide production test [20], nitrate reduction test [21], gelatin liquefaction test [22], sugar or alcohol fermentation tests [18] were performed to determine physiological and biochemical characteristics of the isolated bacteria.

#### 2.9.3. Molecular Identification

Antagonistic bacteria that grew well in NA culture medium were centrifuged at 12,000 rpm for 2 min, the supernatant was discarded, and the cells were collected. The cells were washed three times with sterile water, the cell walls were broken with lysozyme, and the cells were treated with proteinase K. The cells were then extracted with phenol: chloroform isoamyl alcohol at a volume ratio of 25:24:1. The precipitated DNA was dissolved in TE buffer and stored at −20 °C for later use.

The PCR amplification was carried out using 16S rRNA universal primers (27F: 5’-AGAGTTTGATCATGGCTCAG-3’, and 1492R: 5’-TACGGCTACCTTGTTACGACTT-3’) with the extracted total DNA from the antagonist bacteria as the template followed by *gyrB* gene sequencing using universal primers (UP1: 5’-GAAGTCATCATGACCGTTCTGCAYGCNGGNGGNAARTTYGA-3’, and UP2r: 5’-AGCAGGGTACGGATGTGCGAGCCRTCNACRTCNGCRTCNGTCAT-3’) with the extracted total DNA from the antagonist bacteria as the template [23,24]. The PCR reaction volume was 50 μL, consisting of 25 μL of 2× Master Mix, 2 μL of genomic DNA template, 2.5 μL of primers 27F and 1492R, and 18 μL of deionized water. The PCR amplification conditions were as follows: an initial denaturation at 94 °C for 5 min, followed by 35 cycles of denaturation at 94 °C for 40 s, annealing at 55 °C for 30 s, and extension at 72 °C for 60 s, with a final extension at 72 °C for 10 min. The product was then stored at 4 °C. The PCR product was verified by 1% agarose gel electrophoresis and sent to Shanghai Sangon Biotechnology Co., Ltd. for sequencing.

The obtained sequences were submitted to GenBank (Accession No. PX508780) and compared with the sequences registered in GenBank using NCBI primer tool blast program (https://blast.ncbi.nlm.nih.gov/Blast.cgi?PROGRAM=blastn&PAGE_TYPE=BlastSearch&LINK_LOC=blasthome). The model strain sequences with high homology to the test bacterial sequences were selected. The sequences were aligned using ClustalX2.1 software [23,25]. After manual correction, cluster analysis was performed using the neighbor-joining method (NJ) using MEGA7.0 software [26]. The confidence of each branch was verified by bootstrapping 1000 times to construct a phylogenetic tree, thereby determining the taxonomic status of the strains.

### 2.10. Antagonistic Bacterial Fermentation System Research

#### 2.10.1. Seed Solution Preparation

After strain X1 was inoculated onto an LB medium plate and incubated for 48 h, it was inoculated into a 250 mL Erlenmeyer flask containing 50 mL of LB liquid medium and cultured at 28 °C and 180 rpm for an additional 48 h.

#### 2.10.2. Optimization of Fermentation Conditions for Antagonistic Strains

Carbon Source Screening: 2% soluble starch, glucose, sucrose, cornstarch, D-maltose, D-fructose, glycerol, and D-lactose were substituted for the carbon source in the basal culture medium. The mycelial growth rate inhibition assay, a significant method in our study, was used to determine the antimicrobial activity against the gray leaf spot pathogen *P. menezesiana*.

#### 2.10.3. Nitrogen Source Screening

To optimize the culture medium and fermentation parameters, we conducted a series of experiments with thoroughness. The optimal carbon source was used as the carbon source, and the nitrogen source in the culture medium was replaced with 2% peptone, tryptone, casein, yeast extract, beef extract, (NH_4_)_2_SO_4_, KNO_3_, and soybean meal, respectively. Other conditions were the same as above.

#### 2.10.4. Inorganic Salt Screening

For our experiment, we used the optimal carbon and nitrogen sources as the carbon and nitrogen sources. No inorganic salts were added as a control. 0.05% of MgSO_4_·7H_2_O, CaCl_2_, KCl, ZnSO_4_·7H_2_O, MnSO_4_·H_2_O, CuSO_4_, and FeSO_4_·7H_2_O were added to the culture medium, respectively. Other conditions were the same as above.

#### 2.10.5. Multi-Factor Orthogonal Experiment

The optimal carbon source, nitrogen source, and NaH_2_PO_4_·2H_2_O, as well as Na_2_HPO_4_·2H_2_O, were selected as variable factors. The L16(45) orthogonal table was used to optimize the culture medium and determine the optimal ratio of each component within it [27]. Fermentation broths of different concentrations were prepared using carbon source (0.5%, 1%, 2%, 5%), nitrogen source (0.5%, 1%, 2%, 3%), NaH_2_PO_4_·2H_2_O (0.1%, 0.2%, 0.3%, 0.4%), and Na_2_HPO_4_·2H_2_O (0.2%, 0.3%, 0.4%, 0.6%), and fermentation culture was carried out.

The exploration of the optimal fermentation conditions of strain X1 was conducted under comprehensive experimental conditions. These conditions included the volume of liquid, initial pH value, inoculation amount, fermentation time, rotation speed, and temperature. In 250 mL Erlenmeyer flasks, the filling volumes were 30, 50, 70, 90, 120 and 150 mL, respectively; the initial pH values were natural pH, 5.0, 5.5, 6.0, 6.5, 7.0, 7.5, 8.0, 8.5 and 9.0, respectively; the inoculation sizes were 3%, 5%, 7%, 10% and 12%, respectively; the fermentation times were 12, 24, 36, 48, 60, 72, 84 and 96 h, respectively; the rotation speeds were 120, 150, 180, 200 and 220 rpm, respectively; and the temperatures were 20, 24, 28, 32, 36 and 40 °C, respectively.

### 2.11. Data Analysis

All data were analyzed using Excel 2007 and SPSS software (Version 24.0). The one-way analysis of variance was used to analyze the differences in the mean values of each treatment, and the least significant difference (LSD) method was employed for the significance test.

## 3. Results

### 3.1. Screening of Antagonistic Bacteria from Soil and Evaluation of Their Antimicrobial Activity

From a total of 65 bacterial strains isolated from 58 soil samples, strain X1 emerged as a significant player. Its antagonistic effects against three crucial pathogens of coconut were identified using plate contrast and mycelial growth rate assays. The inhibitory rates of strain X1 were determined at different additional ratios (1:10, 1:20, 1:50, 1:100, and 1:200), revealing varying degrees of inhibitory activity against mycelial growth of the three pathogens tested. Even at low additional ratios, the filtrates maintained a moderate inhibitory rate against all three pathogens (Table 1) and (Figure 1). The highest inhibitory activity was observed against the leaf spot pathogen *C*. *oryzae* (Figure 2), with an inhibitory effect exceeding 70% against the crown rot pathogen *C. paradoxa* (Figure 3). These results inspire confidence in the potential of strain X1 in biocontrol.

### 3.2. Identification of Antagonistic Bacteria

Strain X1 was inoculated onto LB solid medium and cultured at 28 °C for 2 days. The colonies, nearly circular, with a smooth, moist, raised surface and irregular margins, were observed and described. Their colors were milky white or light yellow (Figure 3A), opaque, and lacked pigment. Microscopically, the bacteria were carefully examined, revealing their rod-shaped nature, spore-bearing, and Gram-positive (Figure 3B). This thorough morphological and biochemical characterization forms the foundation of our identification process.

The physiological and biochemical identification of the antagonistic strain X1 was conducted, following the guidelines of the Manual of Identification of Common Bacteria and Bergey’s Manual of Bacterial Identification. The identification results, presented in Table 2, provide a robust confirmation of the strain’s identity. The physiological and biochemical characteristics of strain X1 align with those of the genus *Bacillus* and the type of strain of *B*. *velezensis*, instilling confidence in the accuracy of the identification.

To further solidify the taxonomic status and species of strain X1, the results of gyrB gene sequencing, using the primers UP1 and UP2r and cluster analysis, showed that strain X1 clustered with multiple strains of *B*. *velezensis* in a single branch and shared 100% homology, provided compelling evidence that it is *B*. *velezensis*. When combined with morphological, physiological, and biochemical properties, the 16S rDNA and *gyrB* sequence analysis confirmed strain X1 as *B*. *velezensis* (Figure 4) and (Figure 5) respectively.

### 3.3. Optimization of Fermentation Conditions for Strain X1

#### 3.3.1. Optimization of Fermentation Medium

Screening of carbon and nitrogen sources

Glucose, D-maltose, and D-fructose demonstrate superior antimicrobial effects when utilized as carbon sources as illustrated (Figure 6A). The choice of carbon sources does not significantly influence the antimicrobial effect of X1. Glucose, being the most used carbon source, plays a pivotal role in this context. When organic nitrogen sources, such as peptone and yeast extract, are employed, the antimicrobial effect of the fermentation filtrate of strain X1 is significantly enhanced (Figure 6B). The selection of peptone, with the most effective antimicrobial properties, as the nitrogen source for the next experiment, further underscores the importance of glucose in antimicrobial research.

#### 3.3.2. Screening of Inorganic Salts

When ZnSO_4_·7H_2_O and CuSO_4_ were added to the culture medium, the fermentation of X1 was inhibited as shown (Figure 7). However, it’s important to note that there was no significant difference when no other inorganic salts were added or when 0.05% MgSO_4_·7H_2_O, CaCl_2_, and NaCl were added. This reassures us that to simplify the culture medium, no inorganic salts were added. 

Orthogonal test results 

The K value and range (R) value in Table 3 show that the order of influence of the four factors on the antimicrobial effect of strain X1 is: peptone > glucose > NaH_2_PO_4_·2H_2_O > Na_2_HPO_4_·2H_2_O. Peptone content has a significant impact on the antimicrobial rate of the strain. There is no significant difference between 0.3% and 0.4% NaH_2_PO_4_·2H_2_O, so 0.3% is selected with confidence. The results of the orthogonal test and variance analysis indicate that the optimal culture medium formula for strain X1 is A3, B4, C3, D3, specifically 2% glucose, 3% peptone, 0.3% NaH_2_PO_4_·2H_2_O, and 0.4% Na_2_HPO_4_·2H_2_O.

#### 3.3.3. Optimization of Fermentation Conditions

Effect of initial pH

Strain X1 can grow and produce antimicrobial substances within a pH range of 5 to 9, with the best performance at pH 7 to 7.5 as shown in (Figure 8). Its natural pH, around 6.7, yields the second-best antimicrobial rate.

Effect of inoculum size

The inoculum size, ranging from 3% to 12% of the liquid volume, significantly influences X1’s antimicrobial effect (Figure 9). The 5% inoculum size, which yields the highest antimicrobial rate, is therefore identified as the optimal inoculum size, a crucial finding that enlightens our understanding of X1’s behavior.

Effect of Liquid Volume

A liquid volume of 50 to 70 mL in a 250 mL Erlenmeyer flask is optimal for achieving the best antimicrobial effect (Figure 10). This volume ensures the provision of necessary nutrients for the growth of strain X1 and, intriguingly, facilitates the production of the most active antimicrobial substances.

Effect of fermentation time

The X1 strain was cultured in the fermentation medium for 12 to 48 h, the antimicrobial effect increased significantly (Figure 11). After 72 h of culture, the antimicrobial activity reached a stable level, providing a reassuring and confident conclusion that the optimal fermentation time is indeed 72 h.

### 3.4. Key Findings

The rotation speed does not significantly affect the antimicrobial properties of the substances (Figure 12). However, within the range of 120 to 180 rpm, the antimicrobial activity increases with the rotation speed. The activity peaks between 180 and 220 rpm, establishing this as the optimal fermentation speed.

### 3.5. Role of the X1 Strain

The X1 strain played a crucial role in research, exhibiting significant differences in growth and production of antimicrobial substances when cultured at different temperatures (Figure 13). The strain produced antimicrobial substances within the temperature range of 20 to 40 °C. The highest antimicrobial activity of the fermentation broth from strain X1 was observed at 28 °C, leading to the selection of this temperature as the optimal fermentation temperature.

## 4. Discussion

The biocontrol potential of microbial strains, particularly antagonistic bacteria, has been recognized as a promising alternative to traditional chemical pesticides in agriculture. In this study, a total of 65 bacterial strains were isolated from the soils of healthy coconut orchards, identified, and evaluated for their antimicrobial activity against key pathogens affecting coconut crops and among these, strain X1, identified as *B. velezensis* through morphological, physiological, biochemical, and *gyrB* gene sequence analysis, exhibited notable antagonistic activity against coconut crown rot pathogen *C. paradoxa*, gray leaf spot pathogen *P. menezesiana*, and leaf spot pathogen *C*. *oryzae*.

The identification of strain X1 as *Bacillus velezensis* is supported by comprehensive morphological and biochemical characteristics. The colony morphology on LB medium was typical of *Bacillus* species, characterized by smooth, moist, and raised colonies with irregular margins. Microscopically, strain X1 was observed to be rod-shaped, Gram-positive, and spore-bearing, which is consistent with the genus *Bacillus*. These results align with findings from previous studies that highlight *B. velezensis* as an effective biocontrol agent against various plant pathogens [8]. Furthermore, *gyrB* gene sequence analysis confirmed the taxonomic classification of strain X1, which clustered with other *B. velezensis* strains, showing 100% homology. This molecular confirmation is crucial for accurate identification and provides a reliable basis for future applications in biocontrol [24].

The antimicrobial activity of strain X1 was evaluated against three major coconut pathogens, with significant inhibition of mycelial growth observed. The highest inhibitory effect was recorded against the leaf spot pathogen *C*. *oryzae*, followed by the crown rot pathogen *C. paradoxa*. Inhibition rates exceeded 70% in some cases, suggesting the robust antimicrobial properties of strain X1. In another study, 10 strains of B. velezensis were isolated from soil and tested for their antagonistic activity against *Apiospora arundinis* were isolated from soil. Among them, T9 showed an average inhibition rate of 78.8% with a zone of inhibition of 9.1 mm [28]. Additionally, in vitro assays carried out with tomato leaves and fruits, shown that the *B. velezensis* 83 cells formulation had an efficacy of control of *B. cinerea* infection of ∼31% on leaves and ∼89% on fruits [29]. Another similar study found that *B. velezensis* F9 exhibited broad-spectrum antifungal activity against eight plant pathogenic fungi, with inhibition ratios ranging from 62.66% to 88.18%. Furthermore, two pot experiments revealed that the strain exhibited biocontrol efficacy against cucumber wilt, with disease control rates ranging from 42.86% to 67.78% [30].

These results are consistent with previous studies demonstrating the efficacy of *B. velezensis* in controlling various phytopathogens, including fungal pathogens in crops like tomatoes and cucumbers [30]. The consistent inhibition observed at various dilution rates of the fermentation filtrate further strengthens the case for using strain X1 as a viable biocontrol agent in sustainable coconut farming.

The inhibitory effects observed with low concentrations of the filtrate are auspicious, as they suggest that *B. velezensis* X1 could be used effectively at reduced application rates, minimizing potential negative impacts on the environment and non-target organisms [6]. Additionally, the bacterium’s ability to produce bioactive compounds that inhibit pathogen growth points to its potential as a biological pesticide, offering a safe and sustainable alternative to chemical treatments.

The optimization of fermentation conditions was a critical step in maximizing the production of antimicrobial compounds by *B. velezensis* X1. The screening of carbon and nitrogen sources revealed that glucose, maltose, and fructose were all effective carbon sources, with glucose being the most efficient in terms of antimicrobial activity. The choice of nitrogen source also played a significant role, with peptone proving to be the most effective organic nitrogen source for enhancing antimicrobial properties, as observed in other studies with *Bacillus* species [31].

The orthogonal test identified the optimal culture medium as consisting of 2% glucose, 3% peptone, 0.3% NaH_2_PO_4_·2H_2_O, and 0.4% Na_2_HPO_4_·2H_2_O, which provides a balanced nutrient profile to support the growth of *B. velezensis* X1 and the production of bioactive metabolites. We determined that an inoculum size of 5% and a fermentation time of 72 h are optimal for producing antimicrobial compounds. These findings align with the results of similar optimization studies, where optimal fermentation conditions are critical for maximizing the yield of active compounds [31].

Moreover, the temperature range of 20–40 °C and the optimal temperature of 28 °C for the growth and antimicrobial activity of strain X1 further support the suitability of this bacterium for large-scale fermentation in both temperate and tropical environments. The optimal rotation speed of 180–220 rpm is also consistent with the need for adequate aeration during fermentation to enhance the production of bioactive substances [32].

The findings from this study have significant implications for the management of coconut diseases. Coconut palm diseases, particularly those caused by fungal pathogens like *C. paradoxa*, *C*. *oryzae*, and *P. menezesiana*, can lead to substantial yield losses. Traditional chemical control methods often lead to resistance issues and environmental concerns, making biocontrol agents, such as *B. velezensis* X1, an attractive alternative [33,34,35].

The successful identification of strain X1 and its antimicrobial activity against these pathogens suggests that it could be developed as a biocontrol agent for use in coconut orchards. The optimization of fermentation conditions also provides a clear pathway for the large-scale production of this bacterium, which is essential for commercial application [36,37,38]. Furthermore, the low-cost and sustainable nature of this biocontrol approach aligns with the growing demand for eco-friendly agricultural practices.

## 5. Conclusions

In conclusion, *B. velezensis* strain X1 exhibits significant potential as a biocontrol agent against key coconut pathogens. Its identification through a combination of morphological, biochemical, and molecular techniques provides confidence in its taxonomic status. The optimization of fermentation conditions has further elucidated the factors that influence its antimicrobial activity, offering a practical approach for large-scale production. This study provides a solid foundation for the future development of *B. velezensis* X1 as a biological control agent in coconut farming, contributing to the sustainability of coconut cultivation and the reduction of chemical pesticide use.

## Figures and Tables

**Figure 1 microorganisms-13-02640-f001:**
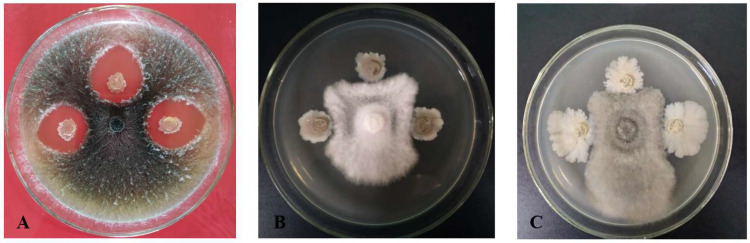
Inhibitory effect of X1 strain on (**A**) *Ceratocystis paradoxa*, (**B**) *Pestalotiopsis menezesiana*, and (**C**) *Curvularia oryzae*.

**Figure 2 microorganisms-13-02640-f002:**
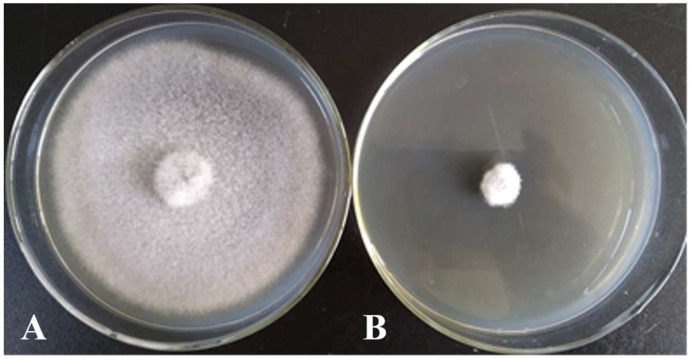
The antimicrobial effect of X1 strain fermentation liquid 1:10 additional ratio on *Curvularia oryzae*. (**A**), control and (**B**), X1 strain.

**Figure 3 microorganisms-13-02640-f003:**
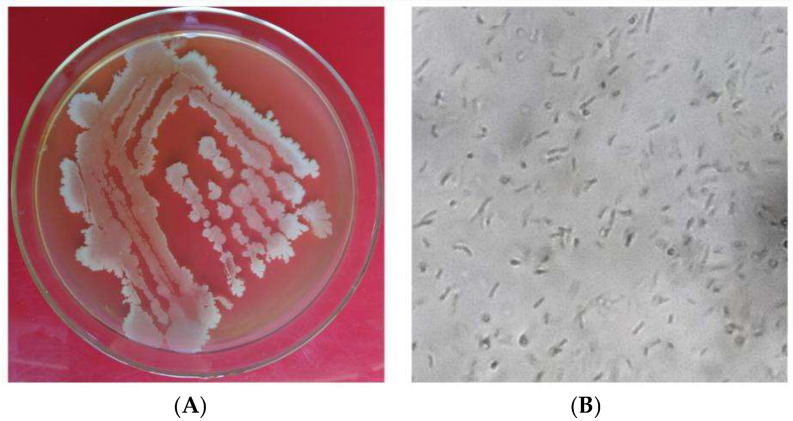
Morphological characteristics of antagonistic strain X1 on LB medium. (**A**), the colonies were nearly circular, with a smooth, moist, and raised surface, irregular edges, and milky white or pale-yellow opaque, and no pigmentation. (**B**), under a microscope, the rod-shaped, Gram-positive bacteria with spores.

**Figure 4 microorganisms-13-02640-f004:**
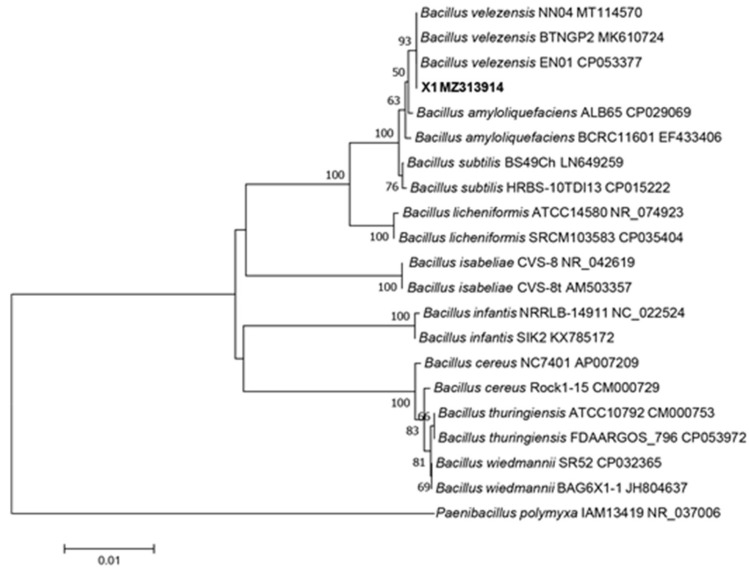
Phylogenetic tree of *Bacillus velezensis* X1 strain (shown in bold) was constructed based on 16S rDNA gene sequences analysis.

**Figure 5 microorganisms-13-02640-f005:**
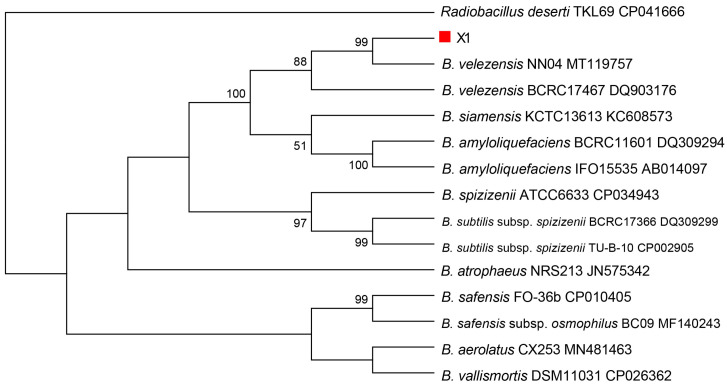
Phylogenetic tree of *Bacillus velezensis* X1 strain (shown in red square) was constructed based on *gyrB* gene sequences analysis.

**Figure 6 microorganisms-13-02640-f006:**
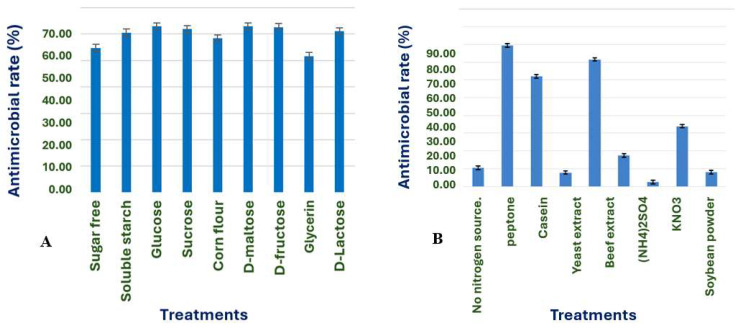
Effects of different carbon and nitrogen sources (**A**,**B**) on the antimicrobial effect of strain X1.

**Figure 7 microorganisms-13-02640-f007:**
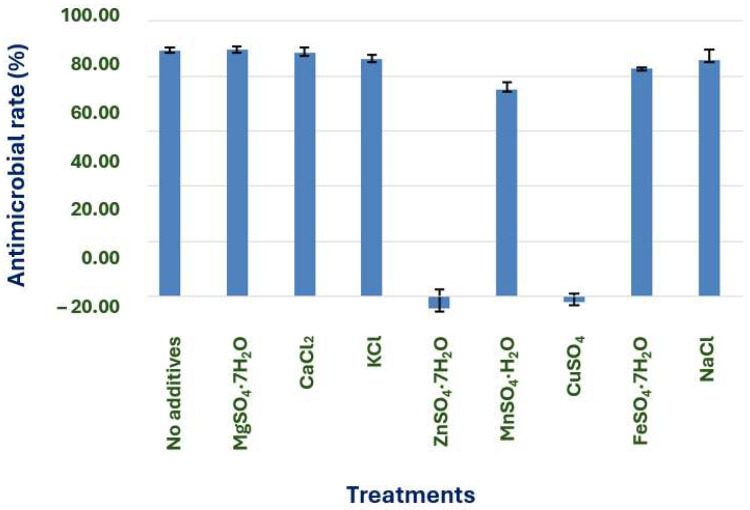
Effects of different inorganic salts on the antimicrobial effect of strain X1.

**Figure 8 microorganisms-13-02640-f008:**
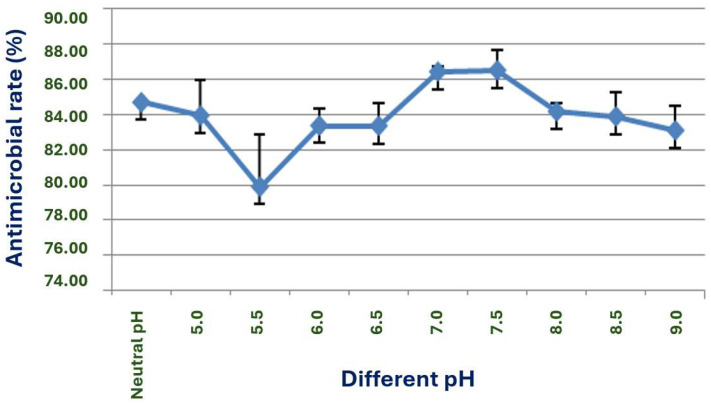
Effect of different initial pH values on the antimicrobial activity of strain X1.

**Figure 9 microorganisms-13-02640-f009:**
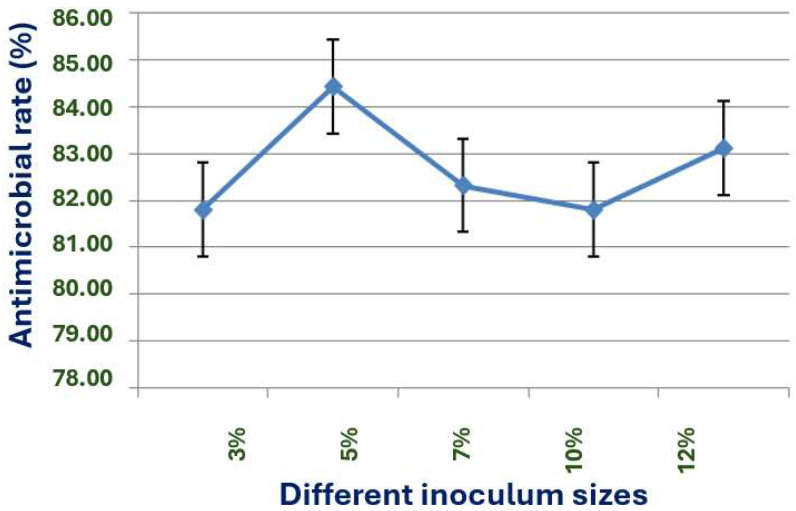
Effect of different inoculum sizes on the antimicrobial effect of strain X1.

**Figure 10 microorganisms-13-02640-f010:**
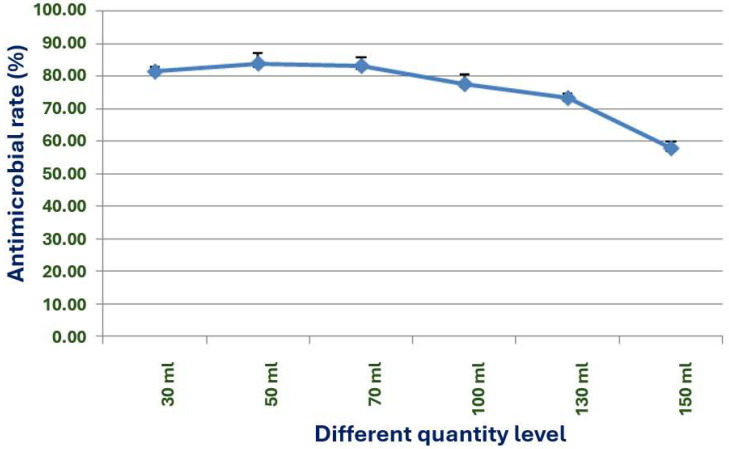
Effects of different bottling volumes on the antimicrobial effect of strain X1.

**Figure 11 microorganisms-13-02640-f011:**
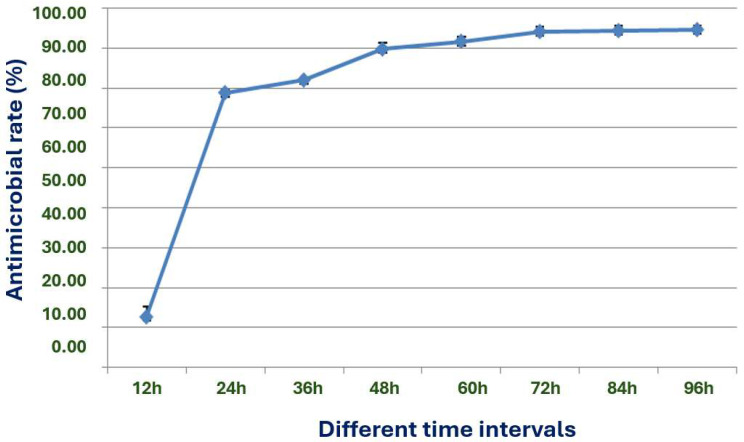
Effects of different fermentation times on the antimicrobial effect of strain X1.

**Figure 12 microorganisms-13-02640-f012:**
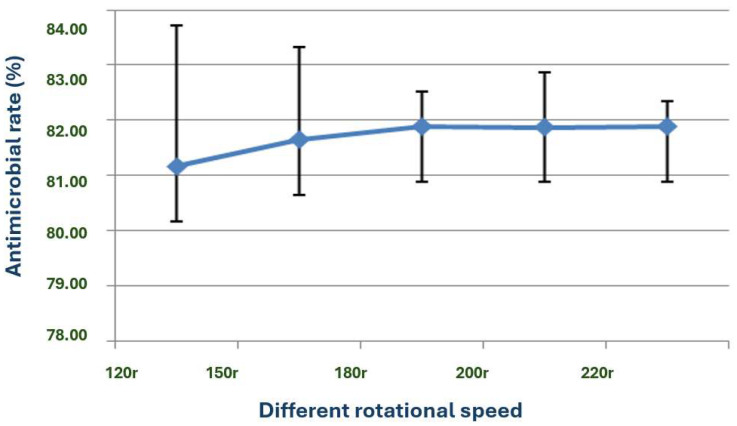
Effects of different shaker speeds on the antimicrobial effect of strain X1.

**Figure 13 microorganisms-13-02640-f013:**
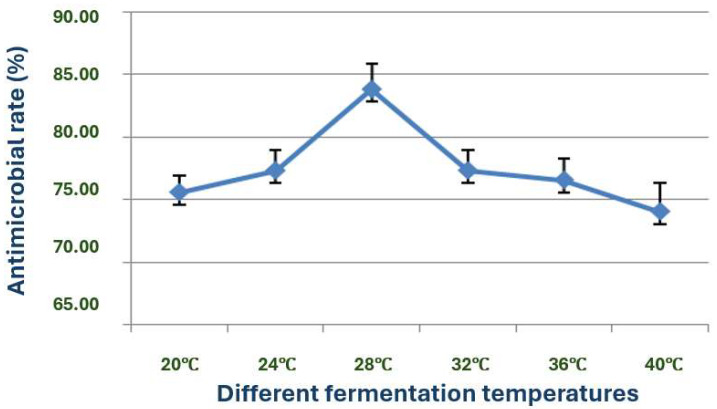
Effects of different fermentation temperatures on the antimicrobial activity of strain X1.

**Table 1 microorganisms-13-02640-t001:** Inhibitory effect of strain X1 on three pathogens at different fermentation filtrate additional ratios.

Pathogens	Additional Volume Ratio				
	1:10	1:20	1:50	1:100	1:200
*Ceratocystis paradoxa*	99.95%	98.92%	95.16%	88.17%	70.97%
*Pestalotiopsis menezesiana*	65.03%	54.78%	34.25%	22.38%	6.85%
*Curvularia oryzae*	90.91%	88.64%	84.66%	81.82%	79.55%

**Table 2 microorganisms-13-02640-t002:** Physiological and biochemical characteristics of strain X1.

Measurement Results	Measurement Results
Methyle red utilization	Galanz dyeing utilization
VP utilization	Hydrogen sulfide utilization
Glucose gas production utilization	Nitrate reduction utilization
No Xylose utilization	Fructose utilization
No Galactose utilization	No Aspartic acid utilization
Glucose utilization	No Lactose utilization
Gelatin liquefaction utilization	Starch hydrolysis utilization
Catalase utilization	No Oxidase utilization
No Sorbitol utilization	No Glycerol utilization

**Table 3 microorganisms-13-02640-t003:** Results of orthogonal experiment for screening the optimal culture conditions for strain X1.

Sr. No.	A	B	C	D	Antimicrobial Rate
1	1	1	1	1	26.66 ± 4.94
2	1	2	2	2	50.60 ± 1.36
3	1	3	3	3	75.44 ± 1.34
4	1	4	4	4	80.17 ± 0.64
5	2	1	2	3	37.25 ± 7.53
6	2	2	1	4	60.63 ± 4.63
7	2	3	4	1	73.97 ± 0.24
8	2	4	3	2	81.96 ± 1.18
9	3	1	3	4	47.92 ± 1.37
10	3	2	4	3	57.69 ± 0.48
11	3	3	1	2	68.64 ± 2.69
12	3	4	2	1	84.62 ± 0.96
13	4	1	4	2	40.83 ± 0.42
14	4	2	3	1	46.73 ± 2.26
15	4	3	2	4	46.73 ± 2.26
16	4	4	1	3	85.80 ± 0.14
k1	58.22	38.17	60.43	57.99	
k2	63.45	53.91	54.80	60.51	
k3	64.72	66.20	63.01	64.05	
k4	55.02	83.14	63.17	58.86	
Very Poor R	9.69	44.97	8.37	6.05	
Order of priority	B > A > C > D				
Excellent	A3	B4	C3/4	D3	

Note: A, B, C, D Carbon source, nitrogen source, NaH_2_PO_4_·2H_2_O and Na_2_HPO_4_·2H_2_O percent content, respectively.

## Data Availability

The original contributions presented in this study are included in the article. Further inquiries can be directed to the corresponding authors.

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
