# Peer review of "Isolation and Identification of Bacteria Bacillus velezensis with Antagonistic Activity Against Major Pathogens of Coconut"

_microorganisms, 2025, doi:10.3390/microorganisms13112640_

Round 1

Reviewer 1 Report

Comments and Suggestions for Authors

Overall comments

In general, the manuscript appears to be interesting for the reader and is quite well written. However, in order to be considered for publication, several points need to be corrected or supplemented:

Introduction

  • L35 - please use italics: Cocos nucifer

  • L40 – please use italics:  Bacillus velezensis, and revise the entire manuscript accordingly.

  • L49–50 – enzymes are not secondary metabolites.

Additionally, please improve the structure of the Introduction section, as it is currently somewhat chaotic. I suggest addressing the issues in the following order:

  1. the importance of coconut,

  2. threats to cultivation, i.e., pathogens,

  3. conventional solutions,

  4. eco-friendly alternatives,

  5. novelty (please elaborate on this point),

  6. aim of the study.

Materials and Methods

  • 2.1. Soil sampling - it would be good if the authors provided the coordinates of the soil sampling site.

  • 2.2. Test pathogens - the source of the pathogens is missing. It is unclear whether these pathogens were isolated as part of the present study or obtained from a collection. Furthermore, it is not explained why the authors selected these three pathogens and referred to them as the main coconut pathogens. The rationale for choosing these particular pathogens should be provided in the Introduction.

  • L112, 118–129 - “1000 ml of distilled water” is missing;
    i) please adopt a single format for describing medium preparation;
    ii) I suggest using the style 17 g ml⁻¹, which seems to better match MDPI requirements.

  • L118 – the composition of Gao liquid medium No. 1 is not provided.

  • Section 2.7 - please specify how strain X1 was applied to the PDA medium: was a pure colony or an inoculum suspension used? (please provide density and volume).

  • L163 - this is probably a mistake; please replace “four pathogens” with “three pathogens.”

  • L185 – in the applied formula, the “%” symbol is missing at the end; it should be 100%.
  • L200 - Are you sure that in this case it is possible to identify your Bacillus isolate at the species level? Wouldn’t additional analyses be required, for example, sequencing of other housekeeping genes such as gyrB, or whole genome sequencing? Wouldn’t it be more appropriate to keep the designation Bacillus sp.?

  • L274-275 – the Methods section does not mention that the fermentation filtrates of strain X1 were tested at different dilution levels (1:10, 1:20, 1:50, 1:100, 1:200).

Results and Discussion

  • L306-307 - this sentence is unnecessary, as it is already described in the last paragraph of section 2.9.3 (Molecular identification).

  • L314, 388, 393, 450 – please replace Bacillus velezensis with B. velezensis.

  • L318 – I suggest changing to D-maltose, D-fructose.

  • Please increase the resolution of Figure 5.

  • 3.3.1 – In the text, the authors state that “The choice of carbon source does not significantly influence the antibacterial effect of X1,” yet they also list glucose, maltose, and fructose as showing the strongest antibacterial activity. Please provide a statistical analysis (e.g., ANOVA with post-hoc test) for these data and for those presented in Figure B.

  • L386 – in section 3.1, the authors state: “From a total of 65 bacterial strains isolated from 58 soil samples, strain X1 emerged as a significant player.” However, in the Discussion, they write: “...three antagonistic bacterial strains were isolated from the soils of healthy coconut orchards...” Please correct this inconsistency in the reported numbers.

  • Please strengthen the discussion section by comparing the obtained inhibition values (70%) with results from other studies on B. velezensis or related species. Example references:

    1. Liao, J., Liang, X., Li, H., Mo, L., Mo, R., Chen, W., ... & Jiang, W. (2023). Biocontrol ability of Bacillus velezensis T9 against Apiospora arundinis causing Apiospora mold on sugarcane. Frontiers in Microbiology, 14, 1314887.

    2. Balderas-Ruíz, K. A., Gómez-Guerrero, C. I., Trujillo-Roldán, M. A., Valdez-Cruz, N. A., Aranda-Ocampo, S., Juárez, A. M., ... & Serrano-Carreón, L. (2021). Bacillus velezensis 83 increases productivity and quality of tomato (Solanum lycopersicum L.): Pre and postharvest assessment. Current Research in Microbial Sciences, 2, 100076.

    3. Spantidos, T. N., Douka, D., Katinakis, P., & Venieraki, A. (2025). Genomic Insights into Plant Growth Promotion and Biocontrol of Bacillus velezensis Amfr20, an Olive Tree Endophyte. Horticulturae, 11(4), 384.

    4. Ta, Y., Fu, S., Liu, H., Zhang, C., He, M., Yu, H., ... & Wang, Y. (2024). Evaluation of Bacillus velezensis F9 for cucumber growth promotion and suppression of Fusarium wilt disease. Microorganisms, 12(9), 1882.

    5. Sendi, Y., Pfeiffer, T., Koch, E., Mhadhbi, H., & Mrabet, M. (2020). Potential of common bean (Phaseolus vulgaris L.) root microbiome in the biocontrol of root rot disease and traits of performance. Journal of Plant Diseases and Protection, 127(4), 453–462.

Reviewer 2 Report

Comments and Suggestions for Authors

The reviewed article is an experimental study devoted to the issue of biological control of fungal infections of coconut plants using a new strain of Bacillus velezensis isolated and identified by the authors. The topic is important and relevant. The Introduction well reveals the research issues. The methods and reagents are described in sufficient detail, but references should be indicated in some places. In the text of the article, the authors should pay attention to the fact that the pathogens under study are fungi, and the antagonistic activity of the strain Bacillus velezensis against them should not be called antibacterial, bacteriostatic, etc., which is incorrect. Table 1 - for all values: Arithmetic mean and error of the arithmetic mean? Statistical significance of differences between options? The captions to the figures should be corrected. The authors use the expression “production of antibacterial compounds” when describing the results of optimizing the nutrient medium, but these indicators were not studied. A number of comments are noted in the text of the article. They should be taken into account and corrected. The Discussion is thorough and in-depth. The Conclusions summarize the results well.

Comments on the Quality of English Language

English should be checked.

Round 2

Reviewer 1 Report

Comments and Suggestions for Authors

I recommend this article in its current form

Author Response

Comment 1: I recommend this article in its current form. 

Response 1: Thank you very much. We are grateful for your valuable suggestions and 
recommendations to improve this research article for publication in Microorganisms. 
Thank you for your time and recommendation.

Reviewer 2 Report

Comments and Suggestions for Authors

The authors have done a lot of work to improve the article, and it has really improved.

The title of subsection 2.8 has not been corrected. “Evaluation of antimicrobial activity against antagonistic bacteria” (Line 201) is incorrect from an English point of view – the illogical construction “against antagonistic bacteria”. It is better to write: Evaluation of antimicrobial activity of antagonistic bacteria.

After correction, the article can be published

Author Response

Comments and Suggestions for authors: The authors have done a lot of work to improve the 
article, and it has really improved. 

Response: Dear Sir, thank you very much for your valuable comments and suggestions to help 
publish this article in Microorganisms. 
We have addressed all your suggestions and comments in the revised version. 

Comment 1: The title of subsection 2.8 has not been corrected. “Evaluation of antimicrobial 
activity against antagonistic bacteria”. (Line 201) is incorrect from an English point of view – 
the illogical construction “against antagonistic bacteria”. It is better to write: Evaluation of 
antimicrobial activity of antagonistic bacteria. 

Response 1: Thank you very much, Sir, for this important correction. We have revised it 
accordingly.
